# miR-965 controls cell proliferation and migration during tissue morphogenesis in the *Drosophila* abdomen

**Pushpa Verma[1], Stephen M Cohen[1,2]\***

[1]Institute of Molecular and Cell Biology, Singapore, Singapore; [2]Department of Cellular and Molecular Medicine, University of Copenhagen, Copenhagen, Denmark

**Abstract** Formation of the *Drosophila* adult abdomen involves a process of tissue replacement in which larval epidermal cells are replaced by adult cells. The progenitors of the adult epidermis are specified during embryogenesis and, unlike the imaginal discs that make up the thoracic and head segments, they remain quiescent during larval development. During pupal development, the abdominal histoblast cells proliferate and migrate to replace the larval epidermis. Here, we provide evidence that the microRNA, *miR-965*, acts via *string* and *wingless* to control histoblast proliferation and migration. Ecdysone signaling downregulates *miR-965* at the onset of pupariation, linking activation of the histoblast nests to the hormonal control of metamorphosis. Replacement of the larval epidermis by adult epidermal progenitors involves regulation of both cell-intrinsic events and cell communication. By regulating both cell proliferation and cell migration, *miR-965* contributes to the robustness of this morphogenetic system.

## Introduction

Tissue morphogenesis is a complex process, through which the organism coordinates cell proliferation and cell death with cell migration and rearrangements to achieve final organ shape and size. Mechanisms controlling these processes play important role in morphogenesis, tissue repair and regeneration, and in cancer (*Friedl and Gilmour, 2009*; *Rorth, 2009*). The abdominal epithelium of *Drosophila* provides a useful model system in which to study the dynamics of tissue morphogenesis in vivo and to explore the genetic and cellular mechanisms that control these complex morphogenetic processes.

During metamorphosis, larval epidermal tissues undergo cellular restructuring and rearrangement to give rise to adult abdominal epithelium. The adult abdominal epithelium is produced from progenitor cells, known as histoblast cells. Histoblasts are small diploid cells, easily distinguishable from the large polyploid larval epidermal cells (LEC) that surround them. Histoblasts are specified in the embryo and lie quiescent throughout larval development (*Guerra et al., 1973*; *Simcox et al., 1991*). There are four pairs of histoblast nests in each segment, which merge to assemble the adult abdominal epidermis (*Madhavan and Schneiderman, 1977*). The anterior and posterior dorsal histoblast nests give rise to the external dorsal cuticle of the abdominal segments (tergites), while the ventral pair give rise to the ventral cuticle (sternites).

During larval stages, histoblasts are arrested in the G2 phase of the cell cycle. At pupariation, an ecdysone pulse triggers the expression of *string* (*cdc25*), which activates cyclin/CDK and pushes the histoblasts into rapid proliferation (*Edgar and O'Farrell, 1990*; *Gautier et al., 1991*). The proliferative phase is divided into two stages with distinct features (*Madhavan and Madhavan, 1980*; *Ninov et al., 2009*). The early 'division phase' is characterized by rapid and synchronous cell division without

*For correspondence: scohen@sund.ku.dk

Competing interests: The authors declare that no competing interests exist.

**eLife digest** Tissues in living organisms are shaped via complex processes that are collectively called 'morphogenesis'. Many researchers have used the fruit fly *Drosophila* as a model to understand morphogenesis, which occurs both during the development of a *Drosophila* embryo and during metamorphosis (when the pupa changes to become an adult fly).

Like other insects, adult fruit flies have three main body sections (a head, a thorax and an abdomen), which are further divided into segments. The adult's abdomen forms inside the pupa from precursor cells called histoblasts. These cells are unusual in that they develop in the embryo but remain inactive during the larval stages of life. During pupation, these cells are reactivated by a hormone called ecdysone, and gradually replace the larva's tissues. However, it was not clear how this process was coordinated.

Verma and Cohen have now demonstrated that a small RNA molecule—a microRNA called *miR-965*—acts in histoblast cells and controls how much these cells divide as well as how they migrate during morphogenesis to replace the larval cells. MicroRNAs regulate other RNAs, called messenger RNAs, typically by targeting them for destruction. This prevents the messenger RNA molecules from being used to make proteins. When flies develop without *miR-965*, they are mostly normal but have defects in the formation of segments in the abdomen.

Verma and Cohen revealed that *miR-965* acts by targeting two important messenger RNAs for destruction. These messenger RNAs encode a protein called String, which regulates histoblast proliferation, and another protein called Wingless. Once the pupa starts to form, the ecdysone hormone reduces the production of *miR-965* to increase histoblast proliferation and migration. The *miR-965* microRNA in turn reduces the level of ecdysone receptor. The ecdysone hormone acts as an all-or-nothing switch to make an irreversible change from the larval to the pupal stage. The hormone boosts its own activity by increasing expression of its own receptor. This 'positive feedback loop' acts like a switch and is very sensitive to small changes in the amount of hormone present.

Verma and Cohen propose that by reducing the levels of the hormone receptor, *miR-965* makes the system more stable. This is because the hormone must first overcome the action of *miR-965* before it can kick off the positive feedback loop. This takes time, and means that any change in the amount of hormone must be around for a while to have an effect. This mechanism buffers against short-lived, small changes in hormone levels that might throw the switch at the wrong time—a feature known as 'robustness'. This seems to be a complicated process to go from one state to another (i.e., from a larva to a pupa). But, the existence of the many distinct checks and balances makes sure the switch is thrown only when it is needed.

---

intervening G2 phases to allow for cell growth. Cells double in number and decrease in size with each division, with little change in the size of the nests. This is followed by a phase of slower division, from 15–40 hr of pupal development, in which proliferation is accompanied by longer intervening gap phases to allow cell growth. During this phase, epidermal growth factor receptor (EGFR) and Insulin receptor/PI3K- signaling coordinate the growth of cells with proliferation (*Ninov et al., 2009*). During the growth phase, the histoblast nests begin to spread to replace the LEC. The expansion of the histoblast nests by cell migration is accompanied by programmed cell death, so that larval cells are replaced by the expanding histoblast population to maintain integrity of the epithelium (*Bischoff and Cseresnyes, 2009*; *Nakajima et al., 2011*).

Patterning of the adult abdominal segments makes use of many of the signaling pathways that pattern the embryonic and larval segments, including *hedgehog*, *wingless*, *decapentaplegic* and EGFR. Interactions between morphogen gradients produced by some of these proteins determine anterior-posterior and dorsal-ventral patterning of segments (*Sanicola et al., 1995*; *Shirras and Couso, 1996*; *Struhl et al., 1997*; *Kopp et al., 1999*; *Ninov et al., 2010*). Although abdominal segmental patterning has been extensively studied, the molecular mechanisms regulating cell division, migration, cell replacement and their interactions during formation of segments remain less well understood. Here, we provide evidence that the microRNA, *miR-965,* is required in the histoblast nests, where it acts via regulation of *string* and *wingless* to control histoblast proliferation and migration during pupal morphogenesis.

# Results

The *miR-965* microRNA is located in the first intron of the *kismet* gene (*Figure 1A*). *kismet* and *miR-965* are transcribed in the same direction. Quantitative PCR showed that a series of P-element insertional mutants near the first exon of the *kismet* gene reduced the level of mature *miR-965* miRNA, suggesting that *kismet* and *miR-965* arise from a single transcription unit (*Figure 1—figure supplement 1*).

To explore the biological functions of *miR-965*, independently from those of *kismet*, we produced targeted deletion mutants to remove the miRNA from the *kismet* intron (*Figure 1A*; [*Chen et al., 2014*]). One mutant allele was made by homologous recombination using the pw25 vector (*Gong and Golic, 2003*). In this allele, the genomic region containing the miRNA hairpin was replaced by a mini-white reporter gene flanked by LoxP sites (*Figure 1A*, w+KO1). The presence of the intron-containing mini-white marker in the *kismet* intron was expected to disrupt *kismet* gene function. Cre-mediated excision of the LoxP-flanked mini-white cassette produced the KO1 allele, in which the miRNA was replaced by a single LoxP site (*Figure 1A*, w-KO1). A second independent deletion was produced using the pRMCE vector, which was designed to allow subsequent retargeting of the locus using Recombination Mediated Cassette Exchange (RMCE) (*Weng et al., 2009*). The miRNA hairpin was replaced by a mini-white marker flanked by inverted attP and LoxP sites. The KO2 allele was made by

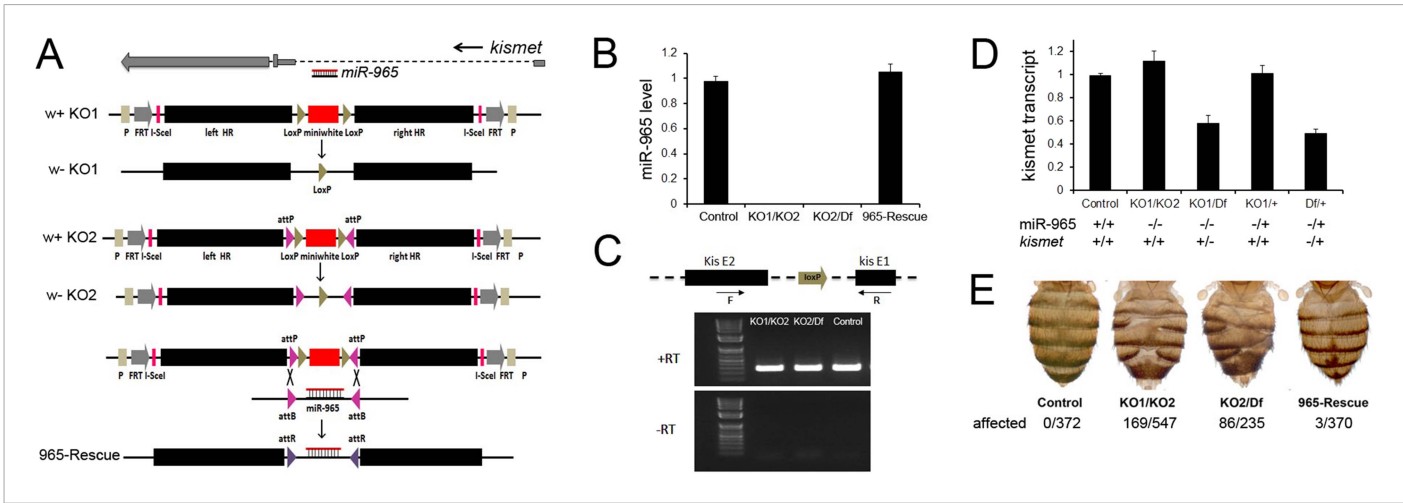

**Figure 1**. The *miR-965* mutant. (**A**) *miR-965* is located within the first intron of the *kismet* gene. The targeting strategies used to produce two independent *miR-965* deletion mutants by ends-out homologous recombination are shown below. Left and Right homology arms cloned into in the targeting vector are shown in black. The w+ KO1 mutant was made by replacing *miR-965* with a mini-white reporter (red) flanked by LoxP sites (grey). Sections are not represented to scale. w-KO1 indicates the targeted allele after Cre-mediated excision of the mini-white cassette. The w+ KO2 mutant was made by replacing *miR-965* with mini-white flanked by LoxP sites and inverted attP sites (pink). w-KO2 indicates the allele after Cre-mediated excision of mini-white. Use of RMCE to replace the mini-white cassette with the miRNA to produce the 965-Rescue allele is shown at bottom. (**B**) *miR-965* RNA level measured by quantitative miRNA PCR. RNA was isolated from adult flies of the indicated genotypes. Control was *w1118*. Df indicates *Df(2L)ED19*. Data represent the average of 3 independent experiments ± standard deviation (SD). (**C**) RT-PCR using primers flanking the first intron of *kismet*. A PCR product of normal size was produced using RNA from flies of each of the indicated genotypes. No product was produced in the absence of reverse transcriptase. (**D**) Quantitative real-time RT-PCR showing kismet transcript levels in *miR-965* mutants (KO1/KO2 and KO1/Df) and the heterozygous KO1/+ and Df/+ controls. Data represent the average of 3 independent experiments ± SD. (**E**) Dorsal aspect of the abdomen from females with the indicated mutant combinations. Control was *w1118*. The number of affected individuals is shown below. ANOVA: p < 0.0001 for each mutant genotype compared to the *w1118* control or to the rescued mutant.

The following figure supplements are available for figure 1:

**Figure supplement 1**. Evidence that *kismet* and *miR-965* arise from a common transcription unit.

**Figure supplement 2**. Phenotype classification.

**Figure supplement 3**. Penetrance of defects in *miR-965* mutants, shown as % of affected individuals.

Cre-mediated excision of the LoxP flanked mini-white reporter (*Figure 1A*, w-KO2). Deletion of the miRNA was verified by quantitative miRNA PCR in animals heteroallelic for the mini-white deleted versions of KO1/KO2 and for the KO2 allele in trans to *Df(2L)ED19*, a chromosomal deletion that uncovers the *kismet* gene (*Figure 1B*). To produce a genetic rescue allele, we used RMCE to replace the mini-white cassette with a fragment containing the miR-965 hairpin (*Figure 1A*, 965-Rescue). *miR-965* expression was restored to normal levels in the 965-Rescue allele (*Figure 1B*).

In the w- versions of the KO1 and KO2 alleles, DNA in the *kismet* intron was replaced by a single LoxP site (*Figure 1A*). To ask whether *kismet* gene function was compromised in these mini-white excised alleles, we performed molecular and genetic tests. *Figure 1C* shows RT-PCR to assess splicing of the first *kismet* intron. A PCR product of normal size was produced in RNA isolated from *miR-965*$^{w-KO1}$/*miR-965*$^{w-KO2}$ and *miR-965*$^{w-KO2}$/Df animals. This qualitative assessment was confirmed by quantitative PCR (*Figure 1D*). *kismet* transcript levels were not reduced in the *miR-965*$^{w-KO1}$/*miR-965*$^{w-KO2}$ flies compared to controls, indicating that splicing of intron 1 was not impaired in these mutants. Genetic complementation tests also showed that the *miR-965*$^{w-KO1}$ and *miR-965*$^{w-KO2}$ mutant alleles were viable in trans to *kismet*[1], a lethal allele, as well as in trans to *Df(2L)ED19*, which removes *kismet*. Together, these data provide evidence that the mini-white excised mutant *miR-965*$^{w-KO1}$ and *miR-965*$^{w-KO2}$ alleles do not compromise *kismet* gene function. For brevity, these mini-white deleted alleles will be referred to as KO1 and KO2 in the text and figures that follow.

## Abdominal segmentation defects

Mutants lacking *miR-965* had only minor effects on survival during development. Mutant adults appeared to be morphologically normal with the exception of defects in abdominal segmentation (*Figure 1E*). Among the affected individuals, the predominant defect was a dorsal gap in one or more abdominal segments, in some cases leading to segment fusion (*Figure 1—figure supplement 2*). In addition, formation of ectopic bristles associated with a polarity defect was observed in ~15% of affected individuals (*Figure 1—figure supplement 3*). Polarity reversal was always accompanied by a gap or segment fusion phenotype. These defects were rescued by restoring miRNA expression using the *miR-965* RMCE rescue allele (*Figure 1E*).

## *miR-965* activity in histoblasts

The abdominal segmentation defects observed in *miR-965* mutants suggested that the miRNA might be required in the histoblasts. To visualize *miR-965* activity, we made use of a sensor transgene consisting of a ubiquitously expressed GFP reporter with a perfect target site for *miR-965* in the 3′ UTR. Sensors of this design allow miRNA activity to be visualized by downregulation of GFP (*Brennecke et al., 2003*). The control sensor, lacking the miRNA-target site, was expressed at comparable levels in the large polyploid LEC and in the smaller histoblast cells (hb, *Figure 2*). *miR-965* sensor GFP levels were lower in the histoblast nests, compared to the adjacent larval cells, particularly in the histoblast cells near the edge of the nests (*Figure 2*). This difference was lost when the sensor was introduced into the *miR-965* mutant background, providing evidence that the reduced GFP level in the histoblasts is due to *miR-965*-mediated repression (*Figure 2*).

## Defects in histoblast development

The *escargot* (*esg*) gene is expressed in histoblasts from the time of their specification in the embryo until the late pupal stage, and provides a marker to visualize histoblast development (*Hayashi et al., 1993*). *Figure 3A* presents still images taken from time-lapse videos of esg-GAL4, UAS-GFP pupae to visualize the first 3 mitotic divisions of the histoblasts. In the controls, cells became smaller and the number of cells doubled after each division (*Video 1*). This pattern of synchronous division was perturbed in the *miR-965* mutant. Asynchronous division of *miR-965* mutant histoblasts resulted in the presence of cells of different sizes (*Figure 3A*, *Video 2*). In the mutant, a subset of histoblast nuclei became pyknotic, fragmented and disappeared, indicating cell death (*Video 3*). Cell death was rare during this phase in the control samples. Synchronous division was restored and cell death was suppressed in the *miR-965*-rescue allele (*Figure 3A*, *Video 4*).

Subsequently, the rate of histoblast nest expansion was slower in the *miR-965* mutant, compared to the controls (*Figure 3B*). Expansion of the histoblast nests was quantified in segments 3 and 4 by monitoring the speed of migration (*Figure 3—figure supplement 1*, *Videos 5, 6*). The average speed

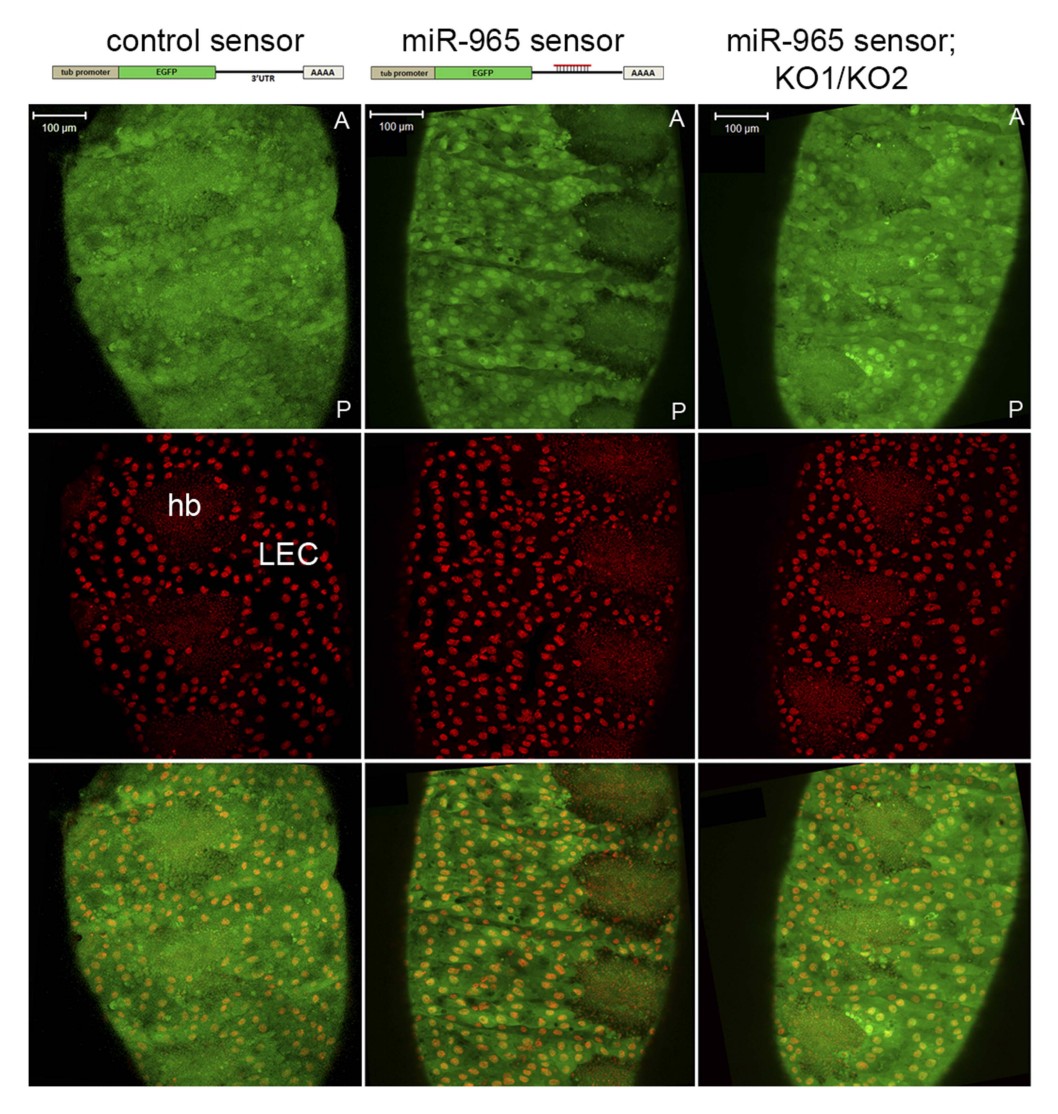

**Figure 2**. *miR-965* expression in histoblasts. Top: design of the control and miR-965 sensor transgenes. EGFP was under control of the tubulin promoter. For the miR-965 sensor, 1 copy of a perfect miR-965 target sequence was placed into the SV40 UTR. Images showing GFP expression from the control sensor (left) and miR-965 sensor (middle) transgenes at 21 hr APF. Histoblast nests consist of small diploid histoblast cells (hb) surrounded by large polyploid larval epidermal cells (LEC). Nuclei were labeled with histone-RFP (red). Downregulation of GFP was lost when the transgene was placed in the KO1/KO2 *miR-965* mutant background (right). Anterior (A), posterior (P). Scale bar: 100 µm.

of migration of third and fourth histoblast nests in the control samples was 15 µm/hr, compare with ~6.5 µm/hr in the mutant. The rate of nest expansion was increased to 12–14 µm/hr by restoring *miR-965* expression with the rescue allele the *miR-965* mutant background (*Video 7*).

LEC generally undergo programmed cell death as the histoblasts nests start to grow (*Nakajima et al., 2011*). In *miR-965* mutants, we observed that some of these cells were still present in the nests, suggesting a failure to eliminate LECs in the mutant (*Figure 3—figure supplement 2*). Delayed expansion of the histoblast nests, combined with the persistence of LECs, is likely to be responsible for the gaps observed in adult abdominal segments.

It was noted that the migration defects observed in the videos seem to be more severe than the segmentation defects observed in the adult flies. However, we did not observe significant pre-eclosion lethality (*Figure 3—figure supplement 3*). Given that there was no evidence for loss of

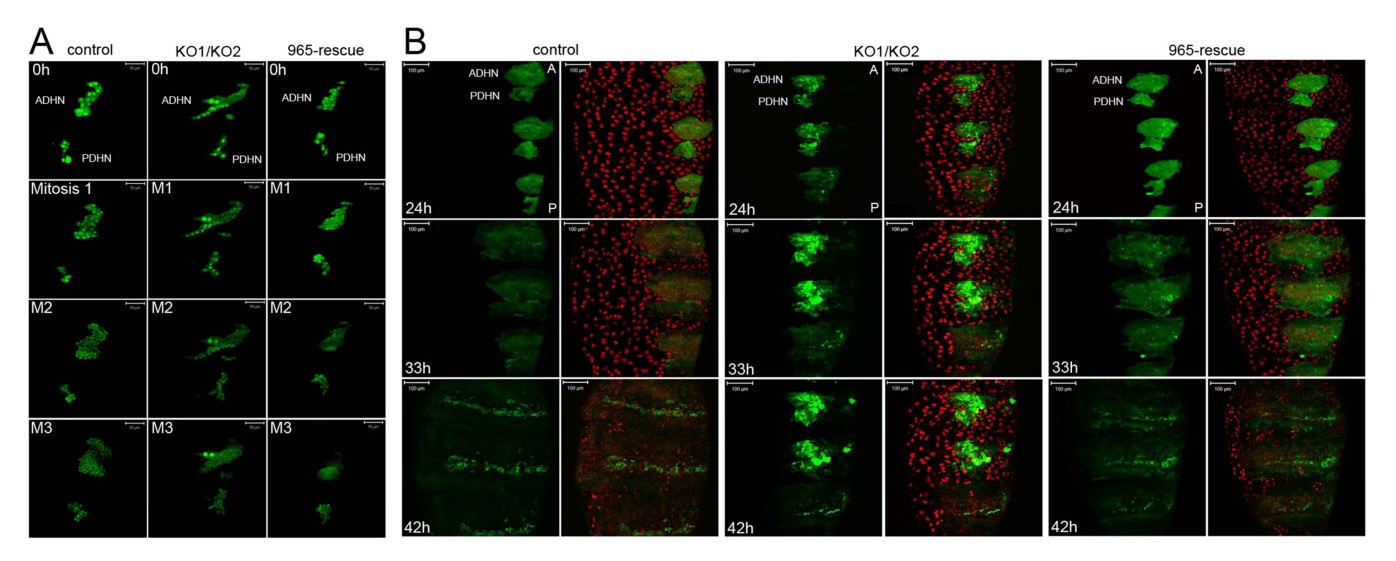

**Figure 3**. Abnormal histoblast proliferation and migration in the *miR-965* mutant. (**A**) Still images taken from time-lapse videos of control, *miR-965* mutant (KO1/KO2) and rescued mutant showing the reduction divisions of the early histoblast proliferation phase. M1, M2 and M3 indicate images taken after mitosis 1, 2 or 3. Imaging was started 0–1 hr APF. Histoblasts were labeled by esg-Gal4 directed expression of UAS-nuclear GFP. ADHN and PDHN represent anterior dorsal histoblast nests and posterior dorsal histoblast nests. Scale bars: 50 μm. Note the different cell sizes in the *miR-965* mutant histoblast nests. (**B**) Still images taken from time-lapse videos at 24, 33 and 42 hr APF from control, *miR-965* mutant and rescued mutant to illustrate expansion of the histoblast nests to replace LECs. Histoblasts were labeled by esg-Gal4 directed expression of cytoplasmic GFP. esg-GAL4 and UAS-GFP were recombined onto the *miR-965* mutant and onto the *miR-965* Rescue chromosome. Nuclei were labeled red with H2-RFP A and P indicate anterior and posterior orientation. Scale bars: 100 μm.

The following figure supplements are available for figure 3:

**Figure supplement 1**. Rate of histoblast nest expansion measured from time-lapse videos.

**Figure supplement 2**. Large polyploid cells in *miR-965* mutant histoblast nests.

**Figure supplement 3**. Pupal survival assays.

a class of more severely affected animals, we suggest that the apparent difference reflects a delay in tissue replacement in the mutant, so that most animals end up with milder defects by the end of pupariation than were apparent during the early pupal time-window in the videos.

### *miR-965* regulates *string* and *wingless*

*miR-965* is predicted to target 69 genes (www.targetscan.org). *polycomb*, *string*, *wingless*, *homothorax*, *Tor*, *Hsp83* and *jumeau* were selected for further analysis based on the quality of the predicted target site and on Flybase annotation suggesting roles in segmentation. Of these, only *string* and *wingless (wg)* mRNAs were upregulated in RNA isolated from *miR-965* mutant pupae, and were also restored to near normal levels in the rescued mutant (***Figure 4A,B***), as would be expected for functional miRNA targets.

The *wg* 3′UTR was used to make a luciferase reporter transgene and tested for regulation by *miR-965* in S2 cells. *miR-965* expression reduced *wg* 3′UTR reporter activity, and this regulation was lost in the *wg* UTR reporter mutated to disrupt pairing with the miRNA seed sequence (***Figure 4C***, mutated residues shown in red). *miR-965* expression also reduced expression of the *string* 3′UTR luciferase reporter (***Figure 4D***). Mutation of the predicted target site to disrupt seed pairing partially offset regulation of the *string* 3′UTR luciferase reporter (***Figure 4D***, mutated residues in red). We noted the presence of a second potential target site nearby (***Figure 4—figure supplement 1***). More extensive mutation to disrupt both sites further compromised regulation by *miR-965* (***Figure 4D***).

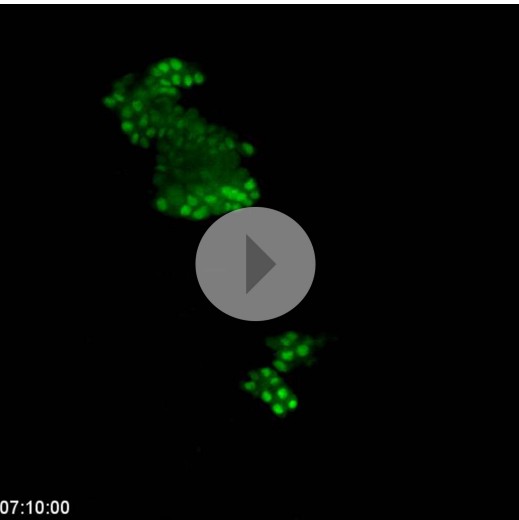

**Video 1.** Control, division phase. Early synchronous divisions of histoblast nests in Esg-GAL4>UAS-nuclear GFP controls. GFP was used to track anterior and posterior dorsal histoblast nests during the early division phase. Animals were collected for imaging at 0 hr APF, the white pre-pupal stage. ADHN: Anterior dorsal histoblast nest. PDHN: posterior dorsal histoblast nest. A and P indicate anterior and posterior orientation. Scale bar: 50 µM. Refers to *Figure 3A*.

**Video 2.** *miR-965* mutant division phase. Early asynchronous divisions in the *miR-965* mutant (KO2, esg-GAL4>UAS-nuclear GFP/KO1). Esg-GAL4>UAS-nuclear GFP was recombined onto *miR-965* RMCE mutant (KO2) chromosome. ADHN and PDHN indicate anterior and posterior dorsal histoblast nests. Animals were collected for imaging at 0 hr APF. Scale bar: 50 µM. Refers to *Figure 3A*.

Other non-canonical target sites might be responsible for the remaining regulation by miR-965, however, we do not exclude the possibility that there could also be indirect effects of *miR-965* on expression of this reporter. These findings provide evidence that miR-965 can regulate expression of *string* and *wg*.

## Contributions of *string* and *wingless* to the miR-965 mutant phenotype

To ask whether the increases in *string* and *wg* expression might be responsible for the *miR-965* mutant phenotype, we first asked if overexpressing them in an otherwise normal genetic background could phenocopy the mutant. esg-Gal4 driven expression of a UAS-string transgene produced abdominal segment gaps, segment fusion and polarity reversal phenotypes, similar to those observed in the *miR-965* mutant (*Figure 5A Figure 5—figure supplement 1*). The proportion of flies with defects caused by *string* overexpression was similar to that in the *miR-965* mutant (*Figure 5B*). *string* overexpression caused asynchronous histoblast division and apoptosis during the early division phase (*Figure 5—figure supplement 2*, *Video 8*). Expansion of the *string*-overexpressing histoblast nests was slowed, and the histoblasts were unable to fully replace the LECs (*Figure 5—figure supplements 2, 3*, *Video 9*), similar to what was observed in *miR-965* mutants. Thus, *string* overexpression was sufficient to reproduce the *miR-965* mutant phenotype.

To assess the contribution of *wg* overexpression, we made use of a UAS transgene directing expression of a temperature sensitive form of Wg protein (*Wilder and Perrimon, 1995*). Use of Wg$^{ts}$ was required to allow stage specific activation of Wg in the histoblasts. Continuous expression of wild-type Wg under esg-Gal4 control was lethal. Wg$^{ts}$ protein is inactive at 25°C. Flies carrying esg-Gal4 and UAS-wg$^{ts}$ were reared at 25°C or shifted to 18°C in the third larval instar to allow expression of active Wg as the histoblasts began proliferation. This resulted in a phenocopy of the segment gap phenotype in 7% of animals (*Figure 5C*). Phenocopy was rare in the animals raised continuously at 25°C to maintain low Wg activity (0.4% affected, *Figure 5C*; p = 0.014 comparing UAS-wg$^{ts}$ at 18° vs 25°, Fisher's exact test).

Next we asked whether limiting target gene overexpression could suppress the mutant phenotype. Lowering *string* activity by introducing *string* mutant alleles reduced the penetrance of the segment gap

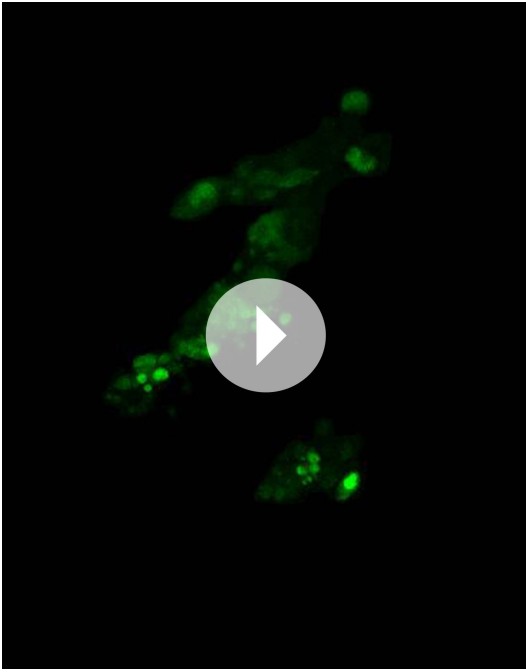

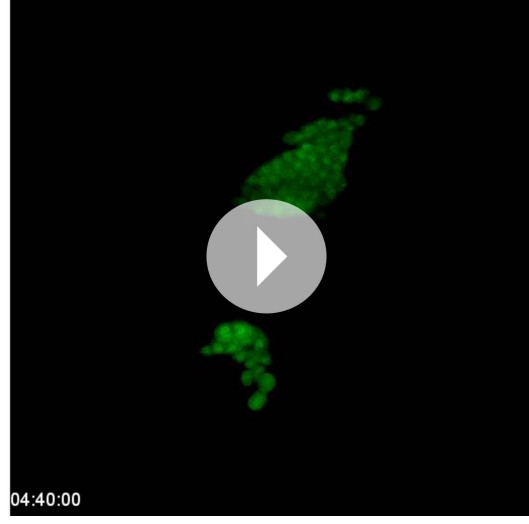

04:40:00

**Video 3.** *miR-965* mutant apoptosis. Apoptotic cells are seen during the early histoblast division phase in the *miR-965* mutant. Esg-GAL4>UAS-nuclear GFP was recombined onto *miR-965* RMCE mutant (KO2) chromosome. ADHN and PDHN indicate anterior and posterior dorsal histoblast nests. Animals were collected for imaging at 0 hr APF. Scale bar: 50 μM. Refers to *Figure 3A*.

**Video 4.** *miR-965*-Rescue division phase. Early synchronous divisions of histoblasts in miR-965 RMCE rescue. Esg-GAL4>UAS-nuclear GFP was recombined onto miR-965 RMCE rescue chromosome. ADHN and PDHN indicate anterior and posterior dorsal histoblast nests. Animals were collected for imaging at 0 hr APF. Scale bar: 50 μM. Refers to *Figure 3A*.

phenotype in the *miR-965* (*KO1/KO2*) mutant background from 25% to 6–7% (*Figure 5D*, *Video 10*, $p < 0.001$ comparing *KO1/KO2* vs *KO1/KO2*; *stg*$^{EY}$ or *stg*$^4$/+, Fisher's exact test). Lowering *string* activity also suppressed the asynchronous division phenotype and the slow cell migration phenotype of the *miR-965* mutant (*Figure 5—figure supplements 4, 5*, *Video 11*).

Wingless protein (Wg) is expressed in the anterior dorsal histoblast nests, during their growth and migration (*Kopp et al., 1999*). The level of Wg protein was higher in the *miR-965* mutant (*Figure 5E*). Wg is a secreted protein, and its distribution appears to be broader, reaching some of the LECs in the mutant, perhaps reflecting the higher level of protein produced (*Figure 5E*). To ask whether this elevated Wg expression contributes to the defects in the mutant, we introduced *wg* mutant alleles into the *miR-965* (*KO1/KO2*) mutant background. The segment gap phenotype was reduced comparing *KO1/KO2* vs *KO1 wg*$^{SP-1}$/*KO2* (*Figure 5F*, $p < 0.05$, Fisher's exact test; suppression by the *wg*$^{I-12}$ allele was not statistically significant). Next, we tested the effect of removing one copy each of *stg* and *wg*. The segment gap phenotype was reduced comparing *KO1/KO2* vs *KO1 wg*$^{SP-1}$/*KO2*; *stg*$^4$/+ (*Figure 5F*, $p < 0.001$, Fisher's exact test). Taking out one copy each of *stg* and *wg* lowered penetrance of the phenotype to ~4%, compared with 6–7% for *stg*/+ alone or 12% for *wg*/+ alone.

These experiments provide evidence that overexpression of *string* and *wg* each contribute to the *miR-965* mutant histoblast phenotypes. The effect of limiting Wg expression in the mutant background may appear to be smaller than that of limiting String. However, to make a meaningful comparison of the relative contribution of these two targets, it would be necessary to restore each of them to normal levels. The genetic method used to reduce target activity in the mutant background does not allow precise control over the final target level, so this question cannot be addressed. We also do not exclude the possibility that there could be other functionally significant targets in addition to Wg and String.

## Ecdysone mediated regulation of *miR-965*

Ecdysone pulses at the beginning of pupariation have been shown to induce *string* expression in order to reactive histoblast proliferation (*Ninov et al., 2009*). In light of the relationship between *miR-965*

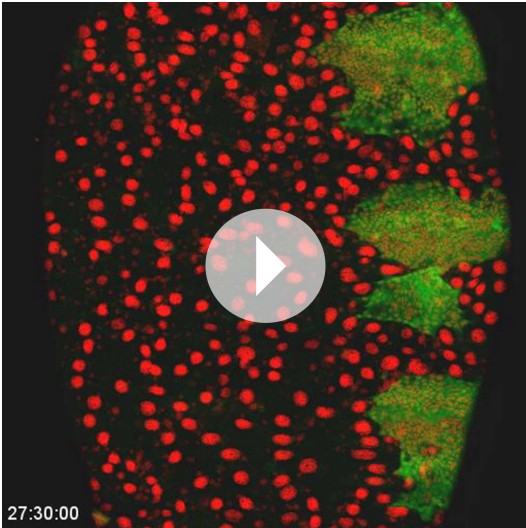

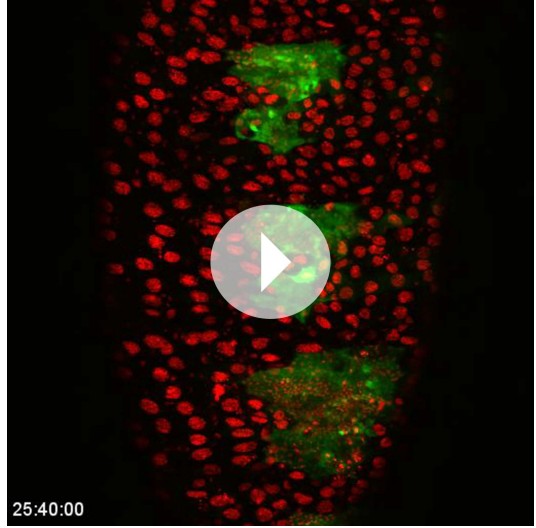

**Video 5.** Control, growth phase. Migration of histoblast nests during the growth phase in an Esg-GAL4>UAS-cytoplasmic GFP control pupa. GFP (green) is used to monitor growth and migration of histoblast nests. H2-RFP (red) marks the nuclei. Big nuclei are LECs and small nuclei are histoblast cells. ADHN and PDHN indicate anterior and posterior dorsal histoblast nests. Scale bar: 100 µM. Refers to *Figure 3—figure supplement 1*.

**Video 6.** *miR-965* mutant growth phase. Delayed migration of histoblast nests during the growth phase in the *miR-965* mutant. Esg-GAL4>UAS-cytoplasmic GFP was recombined onto both KO1 and KO2 mutant chromosomes. This video shows a KO2, esg-GAL4>UAS-cytoplasmic GFP/KO1 pupa. H2-RFP (red) marks the nuclei. ADHN and PDHN indicate anterior and posterior dorsal histoblast nests. Scale bar: 100 µM. Refers to *Figure 3—figure supplement 1*.

and *string*, we asked whether *miR-965* expression might be under Ecdysone control at this stage. We made use of a UAS-EcR^RNAi transgene expressed under esg-Gal4 to reduce Ecdysone receptor levels in the histoblasts. RNAi-mediated depletion of *EcR* mRNA led to an increase in the level of the miR-965 primary transcript and mature miRNA (*Figure 6A*, *Figure 6—figure supplement 1*), and to reduced *string* mRNA levels in RNA samples isolated from early pupae (*Figure 6A*). EcR binding sites have been identified near the host gene, *kismet* (*Gauhar et al., 2009*), consistent with the possibility that ecdysone regulates expression of both *kismet* and *miR-965*. When *miR-965* was overexpressed in the histoblast cells using esg-GAL4, *string* transcript levels were reduced (*Figure 6B*) and histoblast cell divisions were arrested (*Figure 6C*, *Video 12*). This phenotype resembles *EcR-B* mutants, in which histoblast division is compromised (*Bender et al., 1997*). Pupae overexpressing *miR-965* in histoblast cells did not survive beyond ~12 APF, so it was not possible to monitor the later stages of histoblast migration in this genotype.

In light of the finding that EcR limits *miR-965* expression, we asked whether Ecdysone signaling might act via miR-965 to regulate *string*. We introduced a UAS-EcR^RNAi into the *miR-965* mutant background and measured the levels of *string* mRNAs. Depletion of EcR did not reduce *string* mRNA in animals lacking *miR-965* (*Figure 6D*). This provides evidence that Ecdysone signaling is mediated through regulation of the *miR-965* miRNA to regulate histoblast proliferation.

In the course of this analysis, we observed that *EcR* transcript levels increased in the *miR-965* mutant (*Figure 6D*). Both mature and primary transcript levels increased, suggesting an indirect effect of the miRNA on *EcR* transcription. To ask if there might also be a post-transcriptional component to the regulation of *EcR* by miR-965, we used an EcR 3′ UTR reporter transgene linked to GFP (*Varghese and Cohen, 2007*). GFP expression in the histoblast nests did not increase in the miRNA mutant background, indicating indirect regulation of *EcR* by *miR-965* (*Figure 6—figure supplement 2*). These experiments provide evidence for a regulatory feedback relationship between *miR-965* and the Ecdysone receptor (*Figure 6E*). EcR activity limits *miR-965* expression. *miR-965* activity limits *EcR* primary transcript levels, suggesting an effect on transcription.

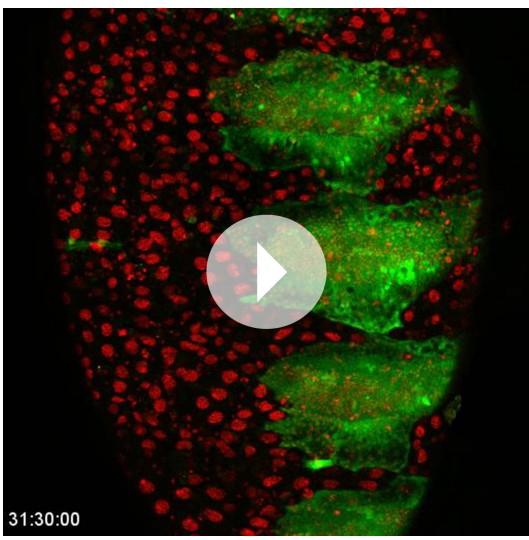

**Video 7.** *miR-965* Rescue growth phase. Migration of histoblast nests during the growth phase in the *miR-965* RMCE rescue genotype. Esg-GAL4>UAS-cytoplasmic GFP was recombined onto the *miR-965* RMCE rescue chromosome. H2-RFP (red) marks the nuclei. ADHN and PDHN indicate anterior and posterior dorsal histoblast nests. Scale bar: 100 µM. Refers to *Figure 3—figure supplement 1*.

## Discussion

Our findings link regulation of the *miR-965* microRNA to the onset of histoblast proliferation at the larval to pupal transition. Previous reports have provided evidence that Ecdysone signaling activates *string* expression to trigger the onset of histoblast proliferation at the beginning of pupal development (*Ninov et al., 2009*). Our findings provide evidence that Ecdysone signaling works though regulation of *miR-965*, which in turn regulates *string*. Interestingly, we also find evidence for negative feedback regulation of *miR-965* on *EcR*. Mutual repression circuitry of this type can contribute a switch-like function: EcR activity lowers *miR-965* activity, which allows greater *EcR* expression/activity by alleviating *miR-965* mediated repression. In a circuit of this design, there will be a delay between reduced transcription of the miRNA primary transcript and the decay of the mature miRNA product. Hence sustained EcR activity is needed to throw the switch.

We have previously reported that *EcR* shows positive transcriptional autoregulation and that this is buffered by *miR-14* in a mutual repression circuit (*Varghese and Cohen, 2007*). Positive feedback allows for a sharp switch-like response, but also makes the system very sensitive to stochastic fluctuation in *EcR* activity. Coupling *EcR* positive auto-feedback to miRNA-mediated repression allows a robust switch function upon Ecdysone stimulation, while protecting the system from the effects of biological noise. This study provides evidence that *miR-965* plays an analogous role in regulating *EcR* response and suggests that *miR-965* confers robustness to the *EcR* response in the histoblasts.

Upregulation of *string* in the *miR-965* mutant contributes to the defects in histoblast proliferation. How misregulation of *string* might contribute to the migration defects is less immediately obvious. Previous work has shown that cell cycle progression in the histoblast population is required to trigger programmed cell death in the surrounding LEC (*Nakajima et al., 2011*). Those authors provided evidence that cell growth and the expansion of the histoblast nests may be required to elicit LEC apoptosis. Although the mechanism by which expansion of the histoblasts triggers LEC death is not clear, elevated *string* expression in the *miR-965* mutant is likely to be responsible for the cell cycle progression defects during this phase, hindering normal LEC removal and histoblast migration.

Persistence of the LECs might also be a consequence of the increased expression of Wg protein in the mutant histoblast nests. Wg acts in combination with EGFR and Dpp signals to control abdominal segment patterning (*Shirras and Couso, 1996*; *Kopp et al., 1999*; *Ninov et al., 2009*, *2010*). These signals are thought to control differential cell adhesion, which may be important for elimination of the LECs as well as for proper segmental fusion of the histoblast nests. Elevated expression of Wg protein may lead to an expanded range of action, perhaps resulting in ectopic Wg activity in the LECs.

Each adult abdominal segment has a well-defined anterior-posterior polarity. Wg is required from 15–20 hr APF for bristle formation and from 18–28 hr APF for tergite differentiation and pigmentation. Overexpression of *wg* has been shown to cause ectopic bristle formation, and *shaggy* mutant clones, which constitutively activate *wg* signaling, can cause polarity reversal in abdominal bristles, while *EGFR*, *FGF*, *dpp* and *Notch* signaling have no effect on the polarity of bristles in adult epidermis (*Lawrence et al., 2002*). Wg levels are normally higher in the posterior region of the anterior histoblast nests and lower more anteriorly. Our finding that Wg levels were elevated and that the distribution of Wg was broader than normal suggests ectopic Wg activity throughout the histoblast

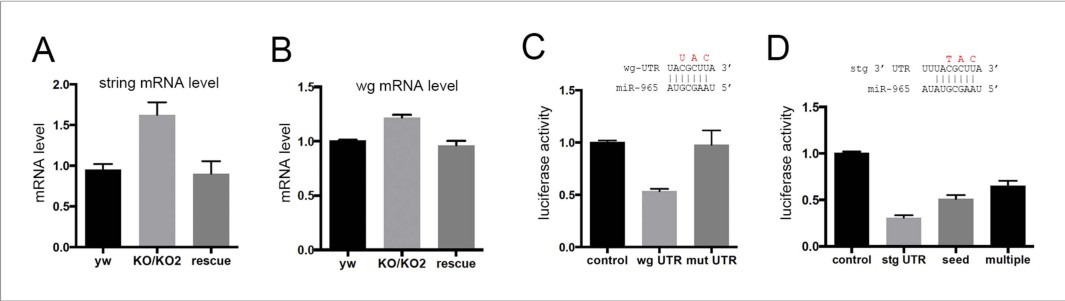

**Figure 4**. miR-965 regulates *string* and *wingless*. (**A**, **B**) string (*stg*) and wingless (*wg*) transcript levels measured by quantitative real time RT-PCR in RNA isolated from $w^{1118}$ control, KO1/KO2 and 965-rescue pupae at 21 hr APF. Data represent the average of three independent RNA collections ± SD. ANOVA: p < 0.01 comparing KO1/KO2 with control or with rescue for *stg* and *wg*. (**C**) Top: diagram of the predicted *miR-965* target site in the *wg* 3′ UTR, showing pairing to the miRNA seed sequence. Residues shown in red were mutated in the mutant version of the *wg* 3′ UTR luciferase reporter. Below: luciferase activity in S2 cells transfected to express a tubulin-promoter *miR-965* transgene, Renilla luciferase and the indicated firefly luciferase reporters. Control indicates the luciferase reporter with the SV40 3′ UTR, which lacks miRNA binding sites. *wg* UTR indicates the intact full-length *wg* 3′ UTR. Mut indicates the *wg* 3′ UTR with the miRNA seed site mutated as indicated in red. Data represent the average of 3 independent experiments ± SD. ANOVA: p < 0.001 comparing control to the intact 3′ UTR. p = 0.001 comparing the intact and site mutant versions of the 3′ UTR. (**D**) Top: diagram of the predicted *miR-965* target site in the *stg* 3′ UTR, showing pairing to the miRNA seed. Residues shown in red were mutated in the seed mutant version of the reporter. The changes made in the extended target site mutant reporter are shown in *Figure 3*. Below: luciferase activity as in panel **C**. Data represent the average of 3 independent experiments ± SD. ANOVA: p < 0.0001 comparing control to the intact 3′ UTR and comparing intact to seed mutant and multiple mutant UTR reporters.

The following figure supplement is available for figure 4:

**Figure supplement 1**. (**A**) Predicted miR-965 sites in the *string* 3′UTR.

nest, including cells that normally experience low Wg levels. Ectopic spread of Wg could be responsible for the formation of ectopic bristles and for the occasional instances of polarity reversal observed in the anterior part of tergites in the *miR-965* mutants.

Replacement of the larval epidermis during metamorphosis involves regulation of both cell-intrinsic events in the abdominal histoblasts and communication between histoblasts and the larval cells they will replace. *miR-965* acts on at least two separate processes required during histoblast morphogenesis. A miRNA with multiple targets can add a layer of regulation, acting across different pathways to integrate their activities (*Herranz and Cohen, 2010*). In doing so, the *miR-965* miRNA appears to contribute to the robustness of this complex morphogenetic system.

## Materials and methods

### Fly strains

$w^{1118}$ was used as the control genotype unless otherwise indicated. *esg-GAL4, UAS-GFP* was obtained from Shigeo Hayashi. *UAS-stg.N4, stg⁴, stg^{EY12388}, wg^{Sp-1}, wg^{I-12}, Kis¹, Kis^{k13416}, Kis^{k11324}, Kis^{k10237}, Kis^{BG01657}, Kis^{KG08532}, Kis^{EY12846}, Df(2L)Exe7702, Df(2L)ED19* are from Bloomington stock center. *EcR-RNAi (v37059)* was from Vienna *Drosophila* Research Center and *P{GawB}esg^{NP7011}* and *ZCL2207 (Atpα-GFP)* was from DGRC, Kyoto.

### Mutant generation

miR-965 (KO1) and miR-965^{RMCE} (KO2) deletion mutants were generated by targeted homologous recombination as described (*Chen et al., 2011, 2014*). Left and right homology arms were amplified by PCR from genomic DNA. The 4055 bp left homology arm was amplified with primers:

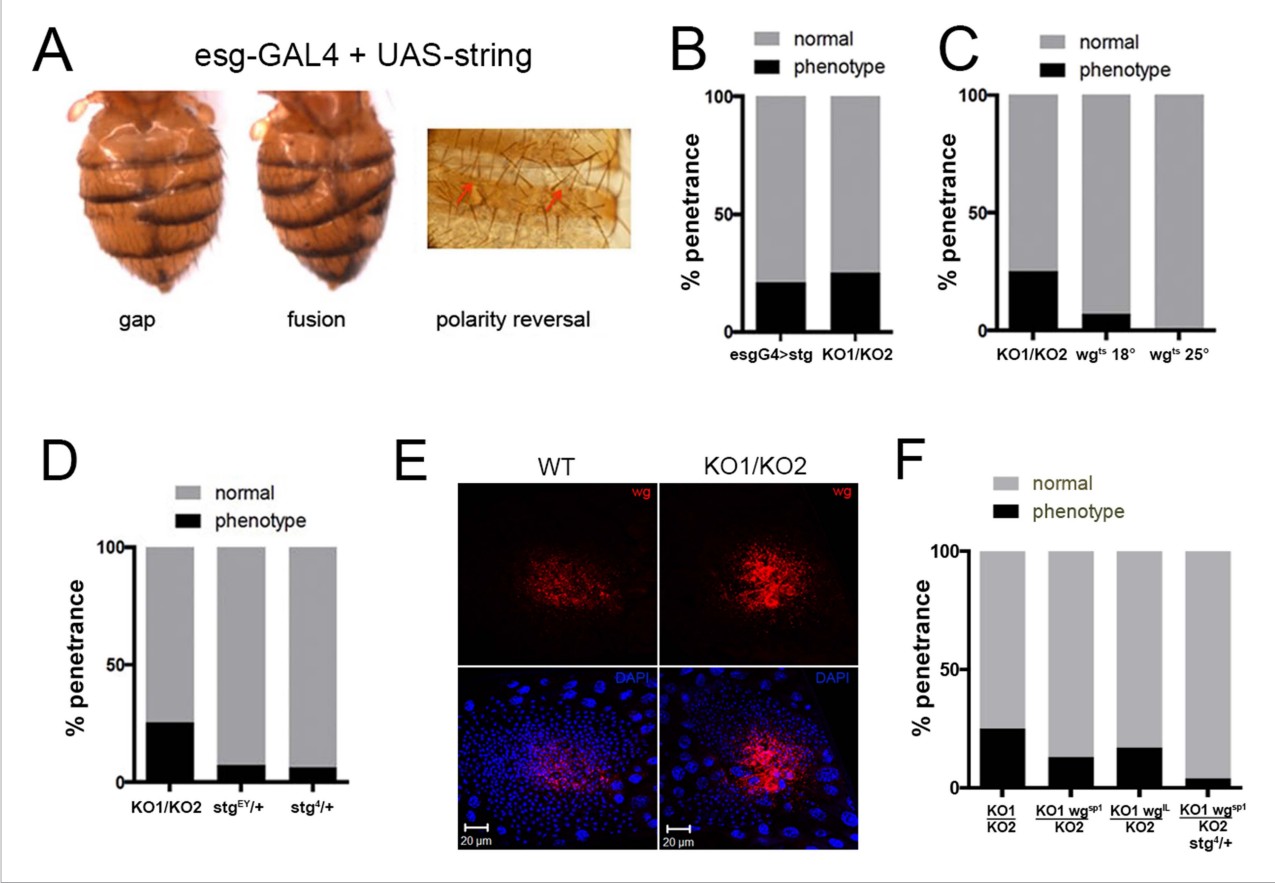

**Figure 5**. Overexpression of *string* and *wg* contributes to the *miR-965* mutant phenotype. (**A**) Dorsal views of abdomens from adult female esg-Gal4 UAS-string flies illustrating the segment gap, segment fusion and polarity reversal phenotypes. (**B**) Penetrance of abdominal defects of all classes in esg-Gal4 UAS-string vs mutant. esg-Gal4 UAS-string: n = 97/469; KO1/KO2 n = 110/446. p = 0.16 Fishers exact test. (**C**) Penetrance of abdominal defects in esg-Gal4 UAS-wg$^{ts}$ flies reared at 18° and 25°C vs KO1/KO2. esg-Gal4 UAS-wg$^{ts}$ reared at 18°C: n = 9/129; esg-Gal4 UAS-wg$^{ts}$ at 25°C n = 1/254; KO1/KO2 n = 110/446. p = 0.014 comparing wg$^{ts}$ at 18 vs 25°C, Fishers exact test. (**D**) Penetrance of abdominal defects comparing KO1/KO2 mutants with KO1/KO2 mutants carrying one copy of *string$^{EY12388}$* or *string$^4$* alleles. p < 0.001 comparing KO1/KO2 to KO1/KO2; stg$^{EY}$/+ or stg$^4$/+ using Fisher's exact test. (**E**) Confocal micrographs showing dorsal histoblast nests of wild-type (WT) and *miR-965* mutant (KO) at ~24 hr APF labeled with anti-Wg (red). Nuclei were labeled with DAPI (blue). Scale bar: 20 μm. Anterior and dorsal histoblast nests in the *miR-965* mutants were not yet fused at 24 hr APF, due to delayed migration. Images were captured using identical microscope settings. (**F**) Penetrance of abdominal segmentation defects comparing KO1/KO2 mutants with KO1/KO2 mutants carrying one copy of *wg$^{SP-1}$* or *wg$^{l-12}$* temperature sensitive alleles or carrying one copy of *wg$^{SP-1}$* and *stg$^4$* together. p < 0.05 comparing KO1/KO2 to KO1, wg$^{SP-1}$/KO2 using Fisher's exact test. KO1/KO2 was not significantly different from KO1, wg$^{l-12}$/KO2, perhaps because *wg$^{l-12}$* is a weaker, temperature sensitive allele. p < 0.001 comparing KO1/KO2 with KO1, wg$^{SP-1}$/KO2; stg$^4$/+ using Fisher's exact test.

The following figure supplements are available for figure 5:

**Figure supplement 1**. The proportion of flies with defects caused by *string* overexpression.

**Figure supplement 2**. Still images from a time-lapse video of esg-Gal4>UAS-string histoblasts.

**Figure supplement 3**. Speed of histoblast nest migration.

**Figure supplement 4**. Rescue of the migration defect of *miR-965* mutants with reduced levels of *string*.

**Figure supplement 5**. Speed of histoblast migration restored by reduced *string* activity.

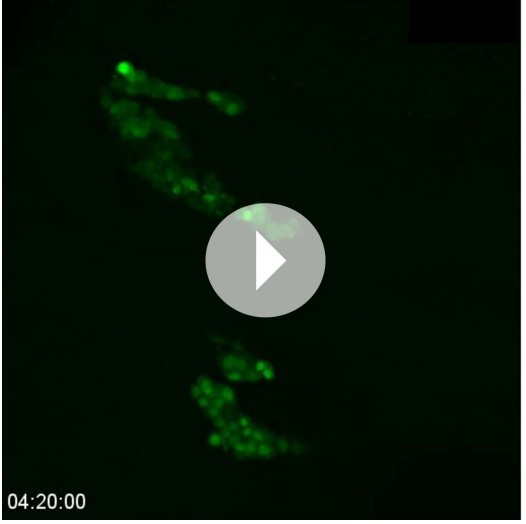

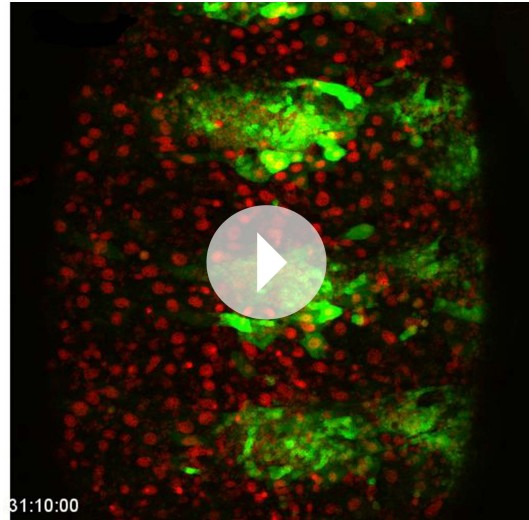

**Video 8.** Esg-Gal4 UAS-Stg division phase. Early divisions in a pupa overexpressing String under esg-GAL4 control (genotype: esg-GAL4, UAS-nuclear GFP/UAS-stg). ADHN and PDHN indicate anterior and posterior dorsal histoblast nests. Animals were collected for imaging at 0 hr APF. Scale bar: 50 μM. Refers to *Figure 5—figure supplement 2*.

**Video 9.** Esg-Gal4 UAS-Stg growth phase. Growth and migration phase in a pupa overexpressing String under esg-GAL4 control, as in *Video 8*. ADHN and PDHN indicate anterior and posterior dorsal histoblast nests. Scale bar: 100 μM. Refers to *Figure 5—figure supplements 2, 3*.

LF- 5′ TTAGAGCTATTGCAACGAAAAGTG 3′, LR- 5′ GTGTAACGGGGATAATAGGATCTG 3′; the 4435 bp right arm was amplified with: RF- 5′ AACACACACAGATGCAGATACAGA 3′ and RR- 5′ AAATAA ACGGTTCACTTCTTCTGC 3′. Following recombination, 153 bp spanning the miRNA-965 hairpin was deleted and replaced with a mini-white cassette. *miR-965* mutants were crossed to heat shock-CRE flies and given heat shock treatment to excise the mini-white cassette. Deletion of *miR-965* in both mutants was confirmed by genomic DNA PCR and by microRNA quantitative-PCR. All genetic tests were done using flies carrying two independently generated alleles or an allele in trans to a chromosomal deletion, *Df(2L)ED19*, which removes the *miR-965* locus.

For generation of Rescue allele, 158 bp of genomic region containing miR-965 hairpin was amplified with primers F- 5′ GCGGGCATGTCGAGGTCGACAAGTAAAATAGCGGAATCAAAATAAT 3′, R- 5′ GC TCTAGAACTAGTGGATCCAACACTTTTCGTTGCAATAGCTC 3′ and replaced *mini-white* gene in *miR-965^{RMCE}* (KO2) mutant allele.

## Sensor and overexpression transgenic generation

miR-965 sensor: for microRNA-GFP sensors, mature miR-965 sequence with primers F- 5′ CTAGAAAGGG GAAAAGCTATACGCTTAC 3′ and R- 5′ TCGAGTAAGCGTATAGCTTTTCCCCTTT 3′ was cloned into 3′UTR of EGFP driven by tubulin promoter in pCasper4.

For UAS constructs, the miR-965 hairpin was amplified using F- 5′ TATAGCGGCCGCAAGTAAAA TAGCGGAATCAAAATAAT 3′ and R- 5′ TATATCTAGAAACACTTTTCGTTGCAATAGCTC 3′ and cloned into pUAST-DsRed.

## 3′UTR reporter assays

miR-965 expression plasmids were generated by cloning the miR-965 hairpin into pCasper-tub-SV40 with primers F- 5′ TCTAGACTTTCATTTTAAGTAAAATAGCGG 3′ and R- 5′ CCTCGAGAACACTTTT CGTTGCAATAGCTCT 3′. For luciferase reporter constructs, 3′UTR of target genes were cloned into pCasper4-tub-Fluc-SV40 firefly luciferase vector.

The following primers were used for cloning 3′UTRs into luciferase vector.

Stg 3′UTR F- 5′ GATCGCCGTGTAATTCTAGAGATGATCGTGCAGTTCGTTATC 3′.

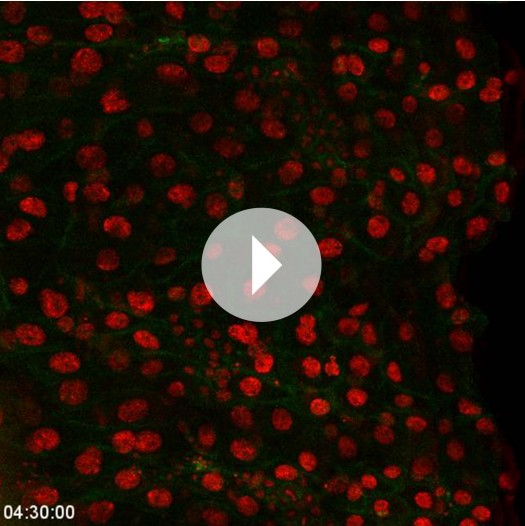

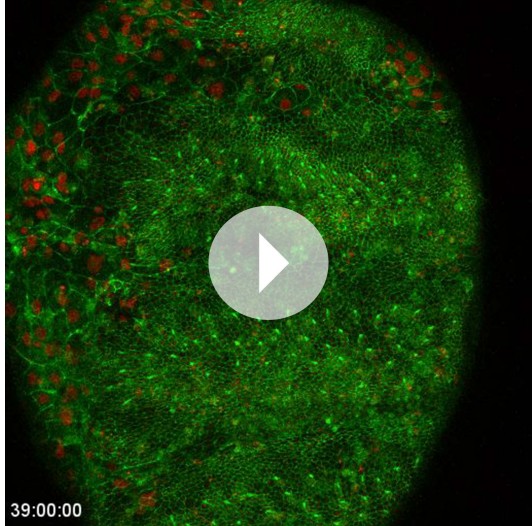

**Video 10.** *miR-965* mutant with reduced string levels division phase. The *string* mutant allele, *stg^EY12388^* was used to reduce *string* levels in the *miR-965* mutant (KO1/KO2) background. *Atpα-GFP* (green) was used to mark cell membranes. H2-RFP (red) marks the nuclei. ADHN and PDHN indicate anterior and posterior dorsal histoblast nests. Scale bar: 50 µM. Refers to *Figure 5D*.

**Video 11.** *miR-965* mutant with reduced string levels with reduced *string levels* growth phase. Normal growth and migration of histoblast nests in the *miR-965* mutant with reduced *string* levels. The *string* mutant allele, *stg^EY12388^* was used to reduce *string* levels in the *miR-965* mutant (KO1/KO2) background. *Atpα-GFP* (green) was used to mark cell membranes. H2-RFP (red) marks the nuclei. ADHN and PDHN indicate anterior and posterior dorsal histoblast nests. Scale bar: 50 µM. Refers to *Figure 5—figure supplements 4, 5*.

 Stg 3′UTR R- 5′ GGCTGCAGGTCGACCTCGAGTTCTTTTTCGTCGTGTATTAATGT 3′.
 Wg 3′UTR F- 5′ GATCGCCGTGTAATTCTAGACCGCCCTCTTCGTTCTTTGT 3′.
 Wg 3′UTR R- 5′ GGCTGCAGGTCGACCTCGAGACTCATTGTCGTTTTGTGTTTTT 3′.
Mutation in the miR-965 seed region in stg 3′UTR was done using following primers:
 Stg mut UTR up F- 5′ GGGCGGAAAGATCGCCGTGTAATTCTAGAGATGATCGTGCAGTTCG 3′.
 Stg mut UTR up R- 5′ CAAATAATGATCATAAATTGTACCTAGCAGAAGTT 3′.
 Stg mut UTR down F- 5′ TTATGATCATTATTTGTTTATTTTTATGTAATCCG 3′.
 Stg mut UTR down F- 5′ ATAAACAAATAAAATTGTACCTAGCAGAAGTT 3′.
 Stg extensive mut 1F- 5′ GATCGCCGTGTAATTCTAGAGATGATCGTGCAGTTCGTTATC 3′.
 Stg extensive mut 1R- 5′ CATCACTTAGGCGTAATGTCGGATAAATAAAGTTTTATGG 3′.
 Stg extensive mut 2F- 5′ACGCCTAAGTGATGCCAGATGTACCCTACTGCTAGGTACAATTTA 3′.
 Stg extensive mut 2R- 5′ GGCTGCAGGTCGACCTCGAGTTCTTTTTCGTCGTGTATTAATGT 3′.
Mutation in miR-965 seed region in wg 3′UTR was done using primers:
 Wg mut up F- 5′ GAACTGCCTGCGTGAGATTCTCGCATGCCAGAGATCCTA 3′.
 Wg mut up R- 5′ CTAATAACAAAGGCTGAGTGGGAGACAAAATACATAACACA 3′.
 Wg mut down F- 5′ TGTGTTATGTATTTTGTCTCCACTCAGCCTTTGTTATTAG 3′.
 Wg mut down R- 5′GGCTGCAGGTCGACCTCGAGACTCATTGTCGTTTTGTGTTTTT 3′.
*Drosophila* Schneider cells (S2) were grown at 25°C in the absence of CO2, with serum free medium (SFM, Gibco) supplemented with L-Glutamine. 2 × 10^6^ S2 cells were transfected in 24-well plates with 25 ng of the firefly luciferase reporter and Renilla luciferase control plasmids, and 250 ng of the miRNA-965 expression plasmid or empty vector. Transfection was done using Cellfectin II (Invitrogen). Transfections were performed in triplicate and each experiment was performed in at least three independent replicates. Cells were lysed 2.5 days after transfection in 100 µl passive lysis buffer, shaken at room temperature for 20 min and dual-luciferase assays were performed according to the manufacturers protocol (Promega).

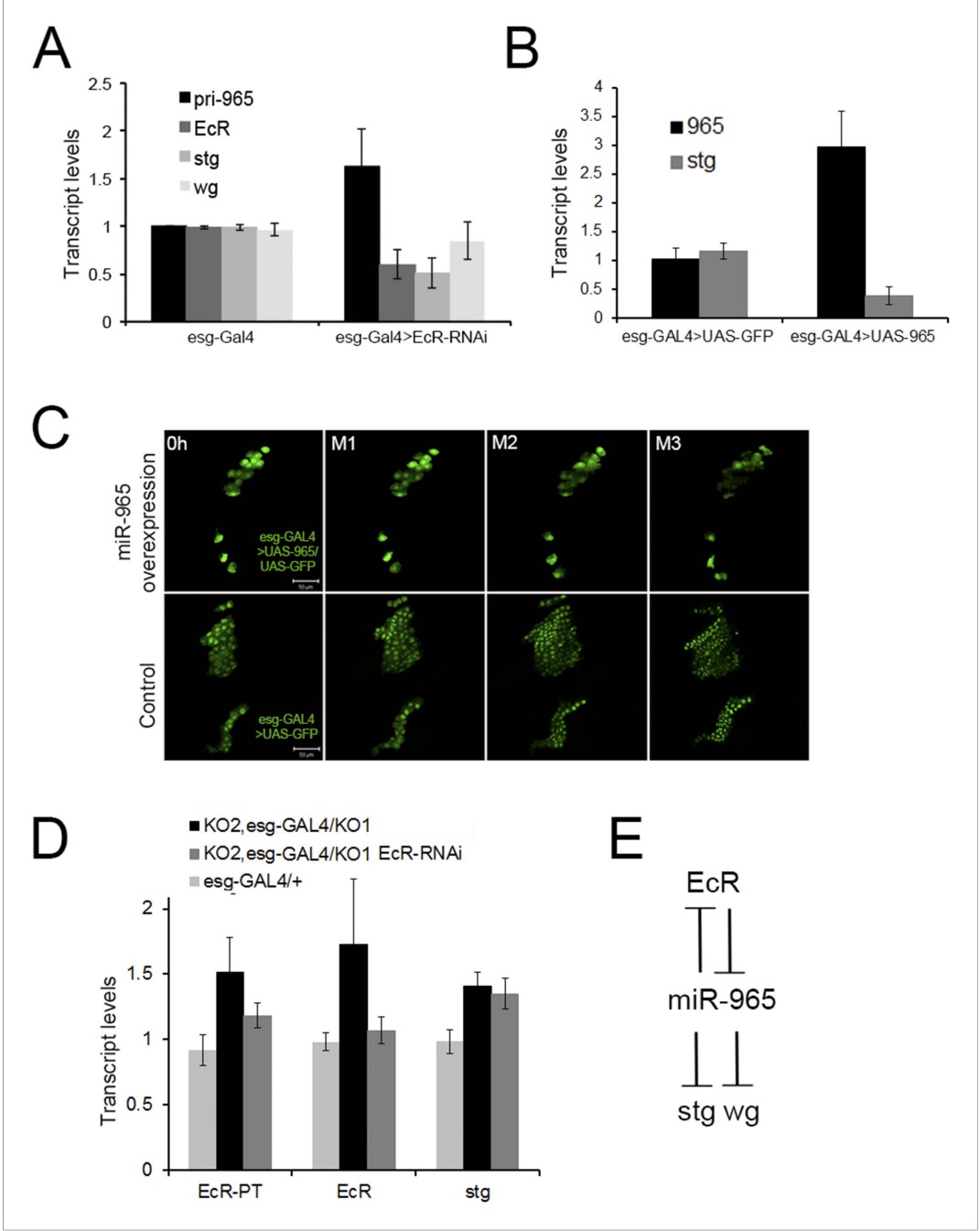

**Figure 6**. Regulation of *miR-965* by ecdysone at the beginning of pupariation. (**A**) Quantitative RT-PCR showing levels of *miR-965* primary transcript, *EcR*, and *string* mRNAs in RNA isolated from pupae expressing esg-GAL4 (control) and esg-GAL4 driving UAS-EcR-RNAi to deplete EcR mRNA. Samples were collected at 0 hr APF. Data were normalized to rp49 and to the esg-GAL4 control. Data represents average of three independent samples ± SD. (**B**) Quantitative RT-PCR showing *string* mRNA in 0 hr pupae overexpressing miR-965 in histoblast cells. For quantitative microRNA PCR, data were normalized to U14, U27, SnoR422. Data were normalized to rp49 for *string* mRNA qPCR. Data represent the average of four independent samples ± SD. (**C**) Images from time-lapse videos showing the effects of *miR-965* overexpression in histoblast cells during the synchronous division phase. M1, M2 and M3 indicate three consecutive mitotic divisions in dorsal histoblast nests. Scale bar: 50 μm. (**D**) Quantitative RT-PCR showing levels of *string*, *EcR* primary transcript (EcR-PT) and mature mRNA in RNA isolated from pupae expressing esg-GAL4 (control), esg-GAL4 in the *miR-965* mutant with and without UAS-EcR-RNAi to deplete EcR mRNA. esg-GAL4 was recombined onto the KO2 mutant chromosome. Samples were collected at 0 hr APF. Data were normalized to rp49 and to the esg-GAL4 control. Data represents average of six independent samples ± SD. p = 0.37 for *stg* levels between KO2, esg-GAL4/KO1 and KO2, esg-GAL4/KO1>EcR-RNAi. p ≤ 0.01 comparing
*Figure 6. continued on next page*

*Figure 6. Continued*

primary and mature *EcR* transcripts between esg-GAL4 control and KO2, esg-GAL4/KO1 mutant samples. (**E**) Diagram of the regulatory relationships between *EcR*, *miR-965* and the miR-965 targets *string* and *wg*. The symbols represent repression of gene expression. *miR-965* and *EcR* repress each other at the primary transcript level. The effect of *miR-965* on *EcR* primary transcript is most likely indirect.

The following figure supplements are available for figure 6:

**Figure supplement 1**. Mature *miR-965* miRNA regulation by EcR.

**Figure supplement 2**. EcR 3′ UTR reporter expression in the *miR-965* mutant.

## Quantitative RT-PCR

Total RNA was extracted from 0 hr or 21 hr pupae and used for cDNA synthesis. Mature *miR-965* transcript level was measured by using TaqMan miRNA assays and normalized to U14, U27 or snoR422 control primers. For target mRNA qRT-PCR, total RNA was treated with RNAse-free DNAse. First strand cDNA was synthesized using oligo-dT primers and SuperScript RT-III (Invitrogen). qRT-PCR was performed using SYBR green (Applied Biosystems). Measurements were normalized to Ribosomal Protein 49. Primers:

rp49 F- GCTAAGCTGTCGCACAAA and rp49 R- TCCGGTGGGCAGCATGTG.
kis F- TTCACGGAAATCATCAAGGA and kis R- CTGTTGCTGTAGCGGATGTG
stg F- ATTCTCCCATTTTCCCAGTTTT and stg R- CTTCCCATCCTATCCTTTCCTT.
wg F- GTCAGGGACGCAAGCATAAT and wg R- GCGAAGGCTCCAGATAGACA.
EcR F- TAACGGCCAACTGATTGTACG and EcR R- GCGGCCAAGACTTTGTTAAGA.
Pri-965 F- AAATCACAAAGCAGAAGAAGTGAA and Pri-965 R- ACAGAAGGGCACATATAACGTACA.

## Live imaging of pupae

For video of the division phase (0–8 hr), 0 hr white pupa were washed with PBS, dried and stuck to imaging dishes (MatTek) with a drop of mineral oil. For growth phase videos, white pupae and allowed to age until 15–20 hr at 25°C. The outer cuticle was removed without damaging internal tissues. Pupae were mounted into imaging dishes with a drop of mineral oil. Zeiss LSM700 and Leica SP5 microscopes were used for imaging and videos were processed by imageJ and photoshop. videos were taken at 5 frames/s for division phase and 10 frames/s for growth phase. For measurement of speed of migration of histoblast nests, the distance moved by the leading edges of the histoblast nests from segment 3 and 4 was measured and migration speed was calculated in micrometer/hour.

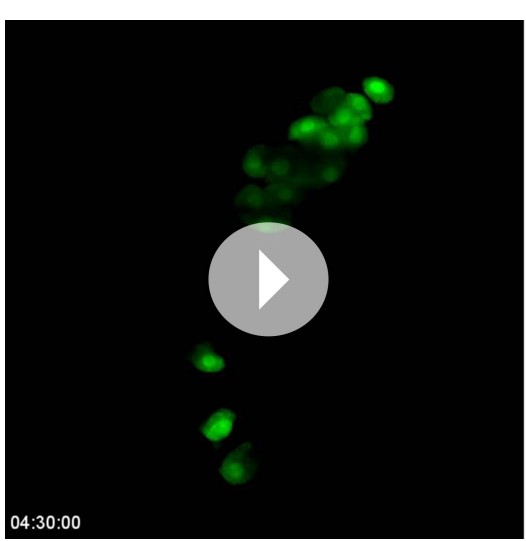

04:30:00

**Video 12.** miR-965 overexpression. Early division arrest in a pupa overexpressing miR-965 under esg-GAL4 control (genotype: esg-GAL4, UAS-nuclear GFP/UAS-miR-965). Animals were collected for imaging at 0 hr APF. Scale bar: 50 μM. Refers to *Figure 6C*.

## Adult cuticle preparation

Adult flies were soaked in Ethanol:Glycerol (3:1), then boiled with 10% KOH at 90°C for 2–5 min. Cuticles were rinsed once with PBT (PBS with 0.1% TritonX-100), then 3 times with PBS, 15 min each and mounted with glycerol.

## Immunohistochemistry

For immunostaining, 0 hr white pupae were transferred to fresh vials and raised at 25°C for staging. Pupae ~24 hr APF were attached to double sided tape and bisected with a blade. Gut

and fat body were removed without disturbing the inner epithelial layer. Cuticle with attached epithelium was washed with PBS and fixed with 4% paraformaldehyde for one and half hour at 4°C. Samples were rinsed 4 × 15 min with PBT (PBS with 0.1% Triton X-100) and blocked for 2 hr with 1% BSA in PBT. Samples were incubated with mouse anti-Wg antibody (1:20, DSHB) overnight at 4°C. Samples were washed with PBT (4 × 15 min) and incubated with AlexaFluor secondary antibody (1:500, Invitrogen) for 2 hr at room temperature. Samples were washed again 4 × 15 min with PBT and stained with DAPI (1:1000) for 10 min at room temperature before mounting.

## Acknowledgements
We thank David Foronda and Adam Cliff for advice on methods.

## Additional information

### Funding

| Funder | Grant reference | Author |
| --- | --- | --- |
| Agency for Science, Technology and Research (A*STAR) | Institute of Molecular and Cell Biology | Pushpa Verma, Stephen M Cohen |
| Novo Nordisk Foundation | NNF12OC0000552 | Stephen M Cohen |

The funders had no role in study design, data collection and interpretation, or the decision to submit the work for publication.

### Author contributions
PV, Conception and design, Acquisition of data, Analysis and interpretation of data, Drafting or revising the article; SMC, Conception and design, Analysis and interpretation of data, Drafting or revising the article

### Author ORCIDs
Stephen M Cohen, http://orcid.org/0000-0003-2858-9163

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
