## [Decision Letter]

Thank you for sending your work entitled “miR-965 controls cell proliferation and migration during tissue morphogenesis in the *Drosophila* abdomen” for consideration at *eLife*. Your article has been favorably evaluated by K VijayRaghavan (Senior Editor), a Reviewing Editor, and three reviewers.

The reviewers agree that this paper addresses an interesting topic and that the regulation of histoblast proliferation and migration by miR-965 is in principle suitable for *eLife*. There were however, some concerns that the conclusions are not fully supported by the current data. The main areas of concern are as follows:

1) There is an apparent discrepancy between the severe defects in pupal histoblast proliferation and spreading, and the rather mild adult defects. This needs clarification: are the adults escapers with a more mild initial phenotype? What are the frequencies of the phenotypes and pre-eclosion lethality? Generally, the phenotypic analysis needs quantification and explanation.

2) Although the effects on stg look robust, the role of miR-965 on wg is less convincing. The effects of loss of miR-965 on wg expression was weak, and reduction of wg was not a very effective suppressor. This needs clarification with further data or, failing that, the conclusion that wg is a significant target of miR-965 in histoblast development needs to be more cautious.

3) Since mutation of the miR-965 site in the stg 3' UTR does not fully rescue the luciferase reporter expression, it is possible that miR-965 might have other indirect effects which might effect stg. This ought to be acknowledged.

4) mir-965 is predicted to target 69 genes, so it should be clarified whether these are the only two targets with functional relevance to the phenotypes observed. This can be addressed by testing whether reduction of both stg and wg together completely rescue the phenotype of mir-965 mutants? At least it needs to be clear whether there are other relevant targets, even if they are unidentified.

5) The role of ecdysone, which is an important physiological element of the model, also needs some further clarification. How functionally important is the ecdysone regulation of stg in histoblasts? Is it dependent on miR-965? And the effect of EcR loss on wg expression was not explored: if wg is a significant player, this is a gap that needs to be addressed.

Overall, we have limited our major comments to issues that the reviewers believe are necessary to support the model, not to extend it. It is acknowledged that this will require some additional data but the consensus is that these are necessary.

---

## [Author Response]

*1) There is an apparent discrepancy between the severe defects in pupal histoblast proliferation and spreading, and the rather mild adult defects. This needs clarification: are the adults escapers with a more mild initial phenotype? What are the frequencies of the phenotypes and pre-eclosion lethality? Generally, the phenotypic analysis needs quantification and explanation*.

We have checked pupal survival rates and see no evidence of pre-eclosion lethality (Figure 7).

Author response image 1.**DOI:**
http://dx.doi.org/10.7554/eLife.07389.037

Pupal survival was assayed for flies of the indicated genotypes. The data present the total number of surviving adults (live) and the total number of dead pupae (dead). There was no significant difference between the mutant and control genotypes used to make the movies:

p=0.67 comparing KO2 esgG4>GFP/+ vs KO2 esgG4>GFP/KO1 (Mann-Whitney test).

p=1 comparing KO1 esgG4>GFP/+ vs KO1 esgG4>GFP/KO2 (Mann-Whitney test).

Given that there was no significant loss of a class of more severely affected animals, we suggest that the apparent difference reflects a delay in tissue replacement in the mutant, so that most animals end up with milder defects by the end of pupariation than were apparent during the time window for making the movies. These data are shown in Figure 3—figure supplement 3 and mentioned in the text.

Quantification: please note that the quantification of the segment gap phenotype was given in Figure 1. The number of affected individuals and total number examined was shown for each genotype. Quantification of the polarity and further detail on the gap phenotype was provided in Figure 1—figure supplement 3.

*2) Although the effects on stg look robust, the role of miR-965 on wg is less convincing. The effects of loss of miR-965 on wg expression was weak, and reduction of wg was not a very effective suppressor. This needs clarification with further data or, failing that, the conclusion that wg is a significant target of miR-965 in histoblast development needs to be more cautious*.

Wg protein level was higher and its range expanded in the mutant (Figure 5). The question is whether that upregulation has an impact on the phenotype. The evidence in Figure 5 says that Wg upregulation does contribute.

The reviewers’ point is based on the magnitude of the effects of manipulating stg and wg activity in the mir-965 mutant background. Because these data are represented in a quantitative manner, it is tempting to interpret the magnitude of the effect as an indication of how “significant” misregulation of the target is to the phenotype. This might be misleading, for the following reason.

In order to be able to compare the relative contribution of each target, we would need to be able to reduce each of them back to normal levels in the mutant background. We lack the means to control target levels so precisely. So, we prefer to limit our interpretation of this class of experiment to a qualitative assessment: Does upregulation of Wg contribute to the mutant phenotype? We believe that the evidence presented answers this question.

In response to the reviewers’ comments, we have changed the text to highlight that the magnitude of the effects of Wg are smaller than those of String. We have added the following text to help the reader think about how to interpret the magnitude differences:

“Next, we tested the effect of removing one copy each of *stg* and *wg.* The segment gap phenotype was reduced comparing *KO1/KO2* vs *KO1 wg*^*SP-1*^*/KO2*; *stg*^*4*^*/+* (Figure 5, p<0.001, Fisher’s exact test). Taking out one copy each of *stg* and *wg* lowered penetrance of the phenotype to ∼4%, compared with 6-7% for Stg/+ alone or 12% for Wg/+ alone.

These experiments provide evidence that overexpression of *string* and *wg* each contribute to the *miR-965* mutant histoblast phenotypes. The effect of limiting Wg expression in the mutant background may appear to be smaller than that of limiting String. However, to make a meaningful comparison of the relative contribution of these two targets, it would be necessary to restore each of them to normal levels. The genetic method used to reduce target activity in the mutant background does not allow precise control over the final target level, so this question cannot be addressed. We do not exclude the possibility that there could be other functionally significant targets in addition to Wg and String.”

*3) Since mutation of the miR-965 site in the stg 3' UTR does not fully rescue the luciferase reporter expression, it is possible that miR-965 might have other indirect effects which might effect stg. This ought to be acknowledged*.

We added: “however, we do not exclude the possibility that there could also be indirect effects of *miR-965* on expression of this reporter. ”

*4) mir-965 is predicted to target 69 genes, so it should be clarified whether these are the only two targets with functional relevance to the phenotypes observed. This can be addressed by testing whether reduction of both stg and wg together completely rescue the phenotype of mir-965 mutants? At least it needs to be clear whether there are other relevant targets, even if they are unidentified*.

Data on the combined effects of lowering stg and wg have been added to Figure 5. Removing both together further suppressed the defect, beyond the level reached by either alone. ∼4% of individuals still showed a phenotype. This could be due to partial offsetting of the increase in target expression (ie residual target excess not allowing complete suppression) or it could be interpreted to mean that there are other targets. Please see the response to point 2 above. The new text highlights the possibility of additional targets.

*5) The role of ecdysone, which is an important physiological element of the model, also needs some further clarification*.

The reviewers raise multiple points. I’ve separated them for clarity.

How functionally important is the ecdysone regulation of stg in histoblasts? Is it dependent on miR-965?

To address this, we asked if depletion of EcR would affect string mRNA levels in the *miR-965* mutant background. String levels were not affected, so it appears that up-regulation of miR-965 is needed for EcR to act on string mRNA at this stage. Data are in the new Figure 6.

*And the effect of EcR loss on wg expression was not explored: if wg is a significant player, this is a gap that needs to be addressed*.

We examined the relationship between EcR and string at the onset of pupariation (0hr white prepupae), when the Ecdysone pulse induces *string* to trigger histoblast proliferation. Wg expression begins in the histoblast nests from 15hr after pupariation, so it is not evident that we should expect the link between EcR and wg to be analogous to that with string. Nonetheless, we did test *wg* transcript in the 0hr prepupal RNA samples used for the string experiment: there was no effect (Figure 8). We did not include the *wg* data in the revised Figure 6, but can do so if the editors request it.

Author response image 2.**DOI:**
http://dx.doi.org/10.7554/eLife.07389.038

New data added on the relationship between EcR and miR965:

While exploring the relationship between EcR and miR-965, we’ve made an interesting new finding: EcR and miR-965 appear to be linked in a mutual repression feedback loop in which EcR and miR-965 repress each other. The new data on repression of EcR by miR965 consists of:

1) qPCR to show that both EcR mRNA levels increase in the miR-965 background. Both mature and primary transcript were affected, suggesting that miR-965 acts indirectly to limit EcR transcription. Data are in the new Figure 6.

2) We also tested for possible direct effects of miR-965 on an EcR 3’ UTR reporter transgene, but did not see evidence for direct regulation of EcR by the miRNA. The data are in Figure 6—figure supplement 2.

Mutual repression circuitry of this type can contribute a switch-like function: EcR activity lowers 965 activity, which allows greater EcR expression/activity by alleviating miR-965 mediated repression. In a circuit of this design, there will be a delay between reduced transcription of the miRNA primary transcript and the decay of the mature miRNA product. Hence sustained EcR activity is needed to throw the switch.

We previously reported that EcR shows positive transcriptional autoregulation and that this is buffered by miR-14 in a mutual repression circuit (Varghese & Cohen, 2007). Positive feedback allows for a sharp switch-like response, but also makes the system very sensitive to stochastic fluctuation in EcR activity. Coupling EcR positive feedback to miRNA-mediated mutual repression allows a robust switch function upon Ecdysone stimulation, while protecting the system from the effects of random fluctuation in EcR levels that might otherwise trigger a feedback response. Our new data provides evidence that miR-965 plays an analogous role in regulating EcR response and suggests that miR-965 confers robustness to the EcR response in the histoblasts.